# MRI-based anatomical characterisation of lower-limb muscles in older women

Erica Montefiori[1,2]* , Barbara M. Kalkman[1,2], William H. Henson[1,2], Margaret A. Paggiosi[2,3], Eugene V. McCloskey[2,3], Claudia Mazzà[1,2]

**1** Department of Mechanical Engineering, University of Sheffield, Sheffield, United Kingdom, **2** INSIGNEO Institute for *in silico* Medicine, University of Sheffield, Sheffield, United Kingdom, **3** Department of Oncology and Metabolism, Centre for Integrated research in Musculoskeletal Ageing, Mellanby Centre for Bone Research, University of Sheffield, Sheffield, United Kingdom

☯ These authors contributed equally to this work.
* e.montefiori@sheffield.ac.uk

**Data Availability Statement:** The data underlying this study are available on Figshare (https://doi.org/10.15131/shef.data.9934055.v1).

## Abstract

The ability of muscles to produce force depends, among others, on their anatomical features and it is altered by ageing-associated weakening. However, a clear characterisation of these features, highly relevant for older individuals, is still lacking. This study hence aimed at characterising muscle volume, length, and physiological cross-sectional area (PCSA) and their variability, between body sides and between individuals, in a group of post-menopausal women. Lower-limb magnetic resonance images were acquired from eleven participants (69 (7) y. o., 66.9 (7.7) kg, 159 (3) cm). Twenty-three muscles were manually segmented from the images and muscle volume, length and PCSA were calculated from this dataset. Personalised maximal isometric force was then calculated using the latter information. The percentage difference between the muscles of the two lower limbs was up to 89% and 22% for volume and length, respectively, and up to 84% for PCSA, with no recognisable pattern associated with limb dominance. Between-subject coefficients of variation reached 36% and 13% for muscle volume and length, respectively. Generally, muscle parameters were similar to previous literature, but volumes were smaller than those from *in-vivo* young adults and slightly higher than *ex-vivo* ones. Maximal isometric force was found to be on average smaller than those obtained from estimates based on linear scaling of *ex-vivo*-based literature values. In conclusion, this study quantified for the first time anatomical asymmetry of lower-limb muscles in older women, suggesting that symmetry should not be assumed in this population. Furthermore, we showed that a scaling approach, widely used in musculoskeletal modelling, leads to an overestimation of the maximal isometric force for most muscles. This heavily questions the validity of this approach for older populations. As a solution, the unique dataset of muscle segmentation made available with this paper could support the development of alternative population-based scaling approaches, together with that of automatic tools for muscle segmentation.

**Funding:** Financial Disclosure: CM and EVMC received funding from the UK Engineering and Physical Sciences Research Council (EPSRC) Grant through the MultiSim and MultiSim2 projects (EP/K03877X/1 and EP/S032940/1, https://epsrc.ukri.org). CM received funding from the National Institute for Health Research (NIHR) Sheffield Biomedical Research Centre (BRC) in Neuroscience IS-BRC-1215-20017, https://www.nihr.ac.uk/). The funders had no role in study design, data collection and analysis, decision to publish, or preparation of the manuscript.The views expressed are those of the authors and not necessarily those of the NHS, the NIHR or the Department of Health and Social Care (DHSC).

**Competing interests:** The authors have declared that no competing interests exist.

## Introduction

The characterisation of the ability of individual muscles to produce force is of particular relevance in older individuals, for whom ageing-associated muscle loss (sarcopenia) can significantly affect the ability of a muscle to produce strength [1–3]. The ability of a muscle to generate force depends on its fibre composition and characteristics, and on its structural and morphological features [4]. The loss of muscle strength at older ages has been explained by a reduction in muscle mass [5], an increase in slower muscle fibres [4], a higher percentage of intramuscular fat, in combination with a smaller physiological cross-sectional area (PCSA) [4,6].

Being able to quantify lower-limb muscle forces during dynamic tasks can help to understand the capability of an individual to control a movement, with relevant application in the prediction of risk of fall and fractures in older people. Musculoskeletal (MSK) models have been increasingly adopted for this purpose [7–10]. These models, however, often rely on a number of assumptions about anatomical features of muscles, neglecting possible variabilities between subjects and within subject (i.e. body asymmetry) which knowingly affect the accuracy of their outputs [11–15]. Additionally, muscle properties such as the maximal isometric force ($F_{max}$) are often derived from cadaver-based dissection studies [16,17], hence neglecting population-specific features like reduction of muscle strength in the elderly.

Estimated muscle forces obtained from MSK models are known to be sensitive to variations in architectural musculotendon parameters [18]. Moderate and muscle-specific sensitivity to $F_{max}$ has been previously reported in young adults, both for generic-scaled [19] and subject-specific models [13]. In these studies, $F_{max}$ was only made to vary within small ranges [19] or proportionally for all muscles [13], and the effects of muscle- or subject-specificity have not been investigated [20]. Additionally, MSK models do not typically account for loss of muscle strength typically associated to ageing [3,21], which can be both subject- and muscle-specific [22].

In an attempt to overcome the above limitations, clinical measurements of muscle strength, such as those from hand-held dynamometer measurements of grip strength, can be integrated in MSK models [23]. These, however, provide an overall indication of strength rather than muscle-specific strength properties. Medical imaging such as Magnetic Resonance Imaging (MRI) has been successfully adopted for deriving individual muscle volume and muscle length through image segmentation [24]. The ratio of muscle volume and length is proportional to the muscle PCSA [25] and hence to the maximal isometric force a muscle can generate [26].

Except for tendon slack length, which cannot yet be quantified with any routine non-invasive techniques, a full characterisation of the muscle parameters based on MRI is certainly feasible [27], but not commonly pursued due to the time and repeatability challenges associated with image processing. As a result, very little is known about specific characteristics of these parameters, especially in older individuals. The main aim of this study was to investigate lower-limb muscle anatomical characteristics, including volume, length, and PCSA, in a group of post-menopausal women. When enough knowledge about individual muscles parameters is available, this could be used to either build population-based statistical models [28] or, as recently proposed by Handsfield et al. [24] for young individuals, establish their relationship with the body mass or length, overcoming the need for segmenting individual muscles in future applications. The second aim of this paper was to verify the suitability of this approach and to provide the community with a fully characterised database including 3D muscle and bone geometries as obtained from lower-limb MRI from a group of post-menopausal women, in the attempt to foster the community efforts towards the development of automatic image processing and modelling tools.

## Methods

### Participants and data acquisition

Eleven post-menopausal women (mean (standard deviation, SD): 69 (7) y. o., 66.9 (7.7) kg, 159 (3) cm) with no movement limitations were recruited by the Metabolic Bone Centre, Northern General Hospital in Sheffield, UK as part of larger studies (Multisim and Multisim 2, EP/K03877X/1 and EP/S032940/1, https://epsrc.ukri.org). Inclusion criteria were having a bone mineral density T-score at the lumbar spine or total hip (whichever was the lower value) less than or equal to -1. Bone mineral density was measured by dual energy x-ray absorptiometry using a Discovery A densitometer (Hologic Inc., Bedford, MA, USA). Exclusion criteria were: body mass index (BMI) <18 or >35, history of or current conditions known to affect bone metabolism and bone mineral density, history of or current neurological disorders, prescription of oral corticosteroids for more than three months within the last year, history of any long term immobilization (>3 months), conditions that prevent the acquisition of musculoskeletal images, use of medications or treatment known to affect bone metabolism other than calcium/vitamin D supplementation and alcohol intake greater than 21 units per week. The study was approved by the East of England—Cambridgeshire and Hertfordshire Research Ethics Committee and the Health Research Authority and was conducted in accordance with the Declaration of Helsinki (October 2000). Written informed consent was obtained from all participants.

During a hospital visit, full lower-limb MRI was collected using a Magnetom Avanto 1.5 T scanner (Siemens, Erlangen Germany). A T1-weighted scanning sequence was used with an echo time of 2.59 ms, a repetition time of 7.64 ms, flip angle of 10 degrees and voxel sizes of 1.1x1.1x5.0 mm for the long bones and 1.1x1.1x3.0 mm for the joints. In this occasion participants' lower-limb dominance was determined asking them "If you kicked a football which foot would you use?" [29].

### Data processing

**Muscle segmentation.** Lower-limb bones were segmented within the MRI scans using Mimics 20.0 (Materialise, Leuven, Belgium). In each limb, 30 muscles were segmented, initially using the automated muscle segmentation toolbox (Mimics Research 20.0, Materialise, Belgium), followed by manual adjustments when necessary. Inter-operator repeatability of the muscle segmentation procedure was assessed by calculating the ratio between SD and mean (referred to as coefficient of variation, CoV) of the muscle volumes ($V_M$) calculated by three different operators on a subset of three participants. According to literature suggestions [30,31], values of CoV can be considered as acceptable when below 10%. Using a conservative approach, for those muscles where inter-operator CoV was higher than 5% we also performed an intra-operator analysis, asking the same operator to repeat the segmentation three times on the same dataset. Following the latter analysis, we discarded all the muscles with non-acceptable repeatability (CoV > 10%). The Psoas major muscle was removed from the repeatability study since it was partially cut off from the MRI field of view in some cases. Similarly, the foot extensors and flexors were not evaluated, since their external boundaries were not identifiable in many of the MRI datasets.

**Calculation of the maximal isometric force.** Two different approaches were used to calculate $F_{max}$. Firstly, a linear scaling of $F_{max}$ based on lower-limb mass [32], which is typically used in MSK models when individual muscle geometries are not available (Lower-limb mass-based scaling, LLMS). Secondly, $F_{max}$ was calculated as a function of muscle PCSA, calculated from individual muscle volumes and length (Volume and length-based scaling, VLS).

In the LLMS approach [32], $F_{max}$ was linearly scaled to the lower-limb mass according to (1):

$$F_{max} = \frac{m_{LL}}{m_{LLGen}} * F_{maxGen} \tag{1}$$

where $m_{LL}$ is the mass of the lower limbs of the subject, calculated as a product of the volume of the lower limbs (estimated from the MRI) and the density of the tissue [33]), $m_{LLGen}$ is the mass of the lower limbs of the generic OpenSim model gait2392 [17] and $F_{maxGen}$ is the default $F_{max}$ of each muscles in the gait2392 model. An equivalent estimate of $F_{max}$ could be theoretically obtained in the absence of MRI by estimating $m_{LL}$ after a scaling procedure (e.g. using the Scaling Tool in OpenSim [34]).

In the VLS approach, muscle segmentations were used to calculate the muscle volume ($V_M$) and the anatomical muscle length ($l_M$) was calculated as the length of the centreline from the 3D muscle segmentation. This was generated as the line connecting the points representing the topological skeleton of each muscle cross section in the 2D MRI slices. A smooth curve was fitted to the centreline using a moving average filter, with the span of the filter being selected individually for each muscle. Values for $l_M$ were then denoted as the arc length of the fitted smoothed curve constituting the centreline of the 3D segmentations. All above computations were performed in MATLAB R2019b (The Mathworks Inc., Natick, MA, USA). $V_M$ and $l_M$ were then used to calculate the muscle PCSA according to (2):

$$PCSA = \frac{V_M}{l_f^o} = \frac{V_M}{k*l_M} \tag{2}$$

where k is the ratio between a muscle optimal fibre length ($l_f^0$) and length, as taken from the literature [25].

Values of $V_M$ and PCSA were compared to those available in the literature for healthy young adults [24] and cadavers [25,27].

$F_{max}$ was calculated as a product of the PCSA described in Eq (1) and the specific tension ($\sigma$ = 61 N/cm$^2$, [16,35]), [26]:

$$F_{max} = \sigma * PCSA. \tag{3}$$

For the Glutei and Adductor magnus, 1/3 of the total $F_{max}$ value was attributed to each of the three bundles constituting the muscle and used for comparison to the values obtained with the LLMS method.

**Statistical analysis.** All variables were tested for normality using the one-sample Kolmogorov-Smirnov test in MATLAB and null hypothesis were then consistently tested using either a student's t test in the case of normally distributed data or a Wilcoxon signed-rank test in the case of non-normally distributed data. To discard the hypothesis of anatomical symmetry, $V_M$, $l_M$ and PCSA of the muscles belonging to the right and left limb were compared. The percentage difference between the values in the right and left limb was also quantified for all the muscles and all the subjects. CoV was calculated for each muscle to quantify the inter-subject variability.

Linear regressions were computed between total lower limb muscle volume ($V_{TOT}$ equal to the sum of the muscles whose segmentation resulted repeatable) and lower-limb mass, body mass, height, and BMI.

The effect of accounting for individual muscle geometry on the calculated $F_{max}$ was quantified by comparing the $F_{max}$ values obtained using the LLMS and VLS approaches. Percentage

difference between $F_{max}$ estimated with the two methods was calculated. Significance level $\alpha$ was set to 0.05 for all statistical tests.

# Results

## Muscle segmentation

The inter-operator analysis provided higher CoV than the intra-operator analysis (Table 1) for all the muscles tested. The Gastrocnemii and Vastus medialis were easily identifiable and led to very high inter-operator repeatability. The Peronei had the worst inter-operator CoV (close to 50%). Even though better results were found for the intra-operator analysis for the Peroneus brevis (CoV = 7.6%), this was not the case for the Peroneus longus (CoV = 10.9%), which was removed from further analysis together with the Gluteus minimus (CoV = 21.6%).

In light of the high inter-operator differences, only muscle segmentations generated by the same single expert operator were used for the following analyses.

## Muscle anatomical parameters

From the dominance test, all participants resulted right limb dominant.

All investigated parameters were not normally distributed; therefore, non-parametric tests were selected for the statistical analysis. An evident intra- and inter-subject variability was

**Table 1. Repeatability of muscle segmentation.**

| Body segments | Muscles | Inter-op CoV | Intra-op CoV | |
|---|---|---|---|---|
| Thigh and gluteal | Iliacus | 8.0 | 2.6 | |
| | Sartorius | 10.2 | 2.0 | CoV ≥ 10% |
| | Gluteus maximus | 7.0 | 2.0 | CoV < 10% |
| | Gluteus medius | 10.6 | 5.3 | CoV < 5% |
| | Gluteus minimus | 14.6 | 21.6 | Not tested |
| | Tensor fasciae latae | 12.4 | 1.1 | |
| | Adductor brevis | 22.8 | 7.5 | |
| | Adductor longus | 17.7 | 6.0 | |
| | Adductor magnus | 5.9 | 3.6 | |
| | Gracilis | 16.1 | 2.7 | |
| | Biceps femoris long head | 7.6 | 4.7 | |
| | Biceps femoris short head | 9.9 | 4.7 | |
| | Semimembranosus | 9.7 | 6.9 | |
| | Semitendinosus | 6.9 | 5.2 | |
| | Rectus femoris | 7.0 | 5.6 | |
| | Vastus intermedius | 6.6 | 1.1 | |
| | Vastus lateralis | 9.8 | 1.2 | |
| | Vastus medialis | 4.2 | - | |
| Calf | Tibialis anterior | 25.3 | 4.2 | |
| | Tibialis posterior | 12.1 | 8.9 | |
| | Gastrocnemius lateralis | 4.6 | - | |
| | Gastrocnemius medialis | 4.5 | - | |
| | Soleus | 8.6 | 5.9 | |
| | Peroneus brevis | 49.4 | 7.6 | |
| | Peroneus longus | 48.2 | 10.9 | |

Inter- and intra-operator coefficient of variation (CoV) for muscle volume calculated by three operators (inter-op) and by one operator over three repetitions (intra-op).

observed for $V_M$ and $l_M$, as depicted by the bar plots in Figs 1 and 2 (individual $V_M$ and $l_M$ values are available as Supplementary material).

The percentage difference of $V_M$ between the two limbs was above 85% for the Gracilis in one subject and for the Rectus femoris in another subject. A significant difference between the two limbs was found for the $V_M$ of the Sartorius, Gluteus maximus, Adductor magnus, and Vastus lateralis, with lower values in the left limb. Between-subject CoV (see Supplementary material) ranged between 14% (Vastus medialis) and 36% (Sartorius).

The percentage difference of $l_M$ between the two limbs was up to 22% (Adductor brevis). A significant difference between the two limbs was observed for the $l_M$ of the Gluteus medius and Vastus lateralis, with lower values in the left limb. Between-subject CoV (see Supplementary material) ranged between 3% (Sartorius) and 13% (Gastrocnemius lateralis).

Mean and SD of the $V_M$ are reported in Table 2 for the sake of comparison with literature data. Overall, our values were higher than dissection-based muscle volumes from elderly cadavers [25] but smaller than muscle volumes from mixed-age cadavers [27] and MRI-based muscle volumes from healthy young adults [24] both of mixed sexes and females only.

Among the tested anthropometric parameters height and lower-limb mass did not significantly correlate with $V_{TOT}$ (Fig 3), whereas BMI and body mass showed significant correlations, with coefficient of determination, $R^2 = 0.44$ ($p = 0.003$) and $R^2 = 0.50$ ($p = 0.004$), respectively.

The percentage difference in PCSA (Fig 4) between right and left limb ranged between -84% for the Gracilis and Rectus femoris (with smaller PCSA in the right limb) and 38% for the Gracilis (with bigger PCSA in the right limb). Only for Gastrocnemius medialis, Gluteus maximus and medius and Soleus between-limb variations were below 20% for all the subjects.

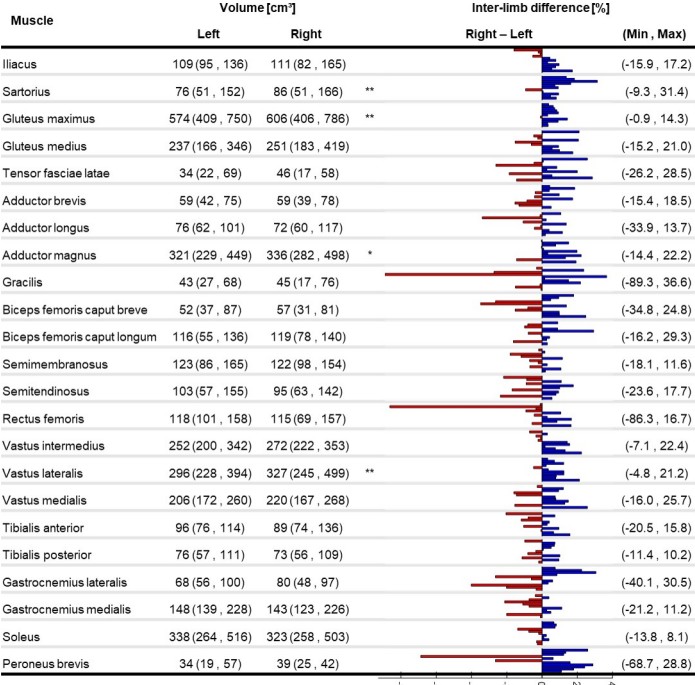

| Muscle | Volume [cm³] | | Inter-limb difference [%] | |
| --- | --- | --- | --- | --- |
| | Left | Right | Right – Left | (Min , Max) |
| Iliacus | 109 (95 , 136) | 111 (82 , 165) | | (-15.9 , 17.2) |
| Sartorius | 76 (51 , 152) | 86 (51 , 166) ** | | (-9.3 , 31.4) |
| Gluteus maximus | 574 (409 , 750) | 606 (406 , 786) ** | | (-0.9 , 14.3) |
| Gluteus medius | 237 (166 , 346) | 251 (183 , 419) | | (-15.2 , 21.0) |
| Tensor fasciae latae | 34 (22 , 69) | 46 (17 , 58) | | (-26.2 , 28.5) |
| Adductor brevis | 59 (42 , 75) | 59 (39 , 78) | | (-15.4 , 18.5) |
| Adductor longus | 76 (62 , 101) | 72 (60 , 117) | | (-33.9 , 13.7) |
| Adductor magnus | 321 (229 , 449) | 336 (282 , 498) * | | (-14.4 , 22.2) |
| Gracilis | 43 (27 , 68) | 45 (17 , 76) | | (-89.3 , 36.6) |
| Biceps femoris caput breve | 52 (37 , 87) | 57 (31 , 81) | | (-34.8 , 24.8) |
| Biceps femoris caput longum | 116 (55 , 136) | 119 (78 , 140) | | (-16.2 , 29.3) |
| Semimembranosus | 123 (86 , 165) | 122 (98 , 154) | | (-18.1 , 11.6) |
| Semitendinosus | 103 (57 , 155) | 95 (63 , 142) | | (-23.6 , 17.7) |
| Rectus femoris | 118 (101 , 158) | 115 (69 , 157) | | (-86.3 , 16.7) |
| Vastus intermedius | 252 (200 , 342) | 272 (222 , 353) | | (-7.1 , 22.4) |
| Vastus lateralis | 296 (228 , 394) | 327 (245 , 499) ** | | (-4.8 , 21.2) |
| Vastus medialis | 206 (172 , 260) | 220 (167 , 268) | | (-16.0 , 25.7) |
| Tibialis anterior | 96 (76 , 114) | 89 (74 , 136) | | (-20.5 , 15.8) |
| Tibialis posterior | 76 (57 , 111) | 73 (56 , 109) | | (-11.4 , 10.2) |
| Gastrocnemius lateralis | 68 (56 , 100) | 80 (48 , 97) | | (-40.1 , 30.5) |
| Gastrocnemius medialis | 148 (139 , 228) | 143 (123 , 226) | | (-21.2 , 11.2) |
| Soleus | 338 (264 , 516) | 323 (258 , 503) | | (-13.8 , 8.1) |
| Peroneus brevis | 34 (19 , 57) | 39 (25 , 42) | | (-68.7 , 28.8) |

**Fig 1. Muscle volume variability.** Median (minimum, maximum) of muscle volume for the right and left limb (significant difference between limbs: * p<0.05, **p<0.01). Individual percentage difference between the limbs is reported as a bar plot where each bar represents a participant: blue positive (red negative) values show that the right leg is bigger (smaller). Minimum and maximum percentage difference across the subjects is reported for each muscle.

| Muscle | Length [cm] | | | Inter-limb difference[%] | |
| --- | --- | --- | --- | --- | --- |
| | Left | Right | | Right – Left | (Min , Max) |
| Iliacus | 33.3 (32.0 , 36.5) | 33.8 (30.9 , 34.9) | | | (-15.9 , 17.2) |
| Sartorius | 59.0 (56.2 , 61.7) | 59.2 (56.6 , 62.6) | ** | | (-9.3 , 31.4) |
| Gluteus maximus | 35.3 (30.3 , 37.3) | 34.3 (28.2 , 35.4) | ** | | (-0.9 , 14.3) |
| Gluteus medius | 28.0 (27.1 , 29.4) | 26.9 (25.4 , 28.3) | | | (-15.2 , 21.0) |
| Tensor fasciae latae | 18.3 (16.3 , 21.7) | 18.8 (16.6 , 21.7) | | | (-26.2 , 28.5) |
| Adductor brevis | 20.2 (18.2 , 23.5) | 20.4 (17.1 , 25.8) | | | (-15.4 , 18.5) |
| Adductor longus | 26.3 (23.2 , 29.5) | 26.6 (21.2 , 29.7) | | | (-33.9 , 13.7) |
| Adductor magnus | 36.3 (29.7 , 38.9) | 35.7 (27.9 , 39.6) | * | | (-14.4 , 22.2) |
| Gracilis | 34.3 (32.2 , 38.8) | 35.7 (31.2 , 39.9) | | | (-89.3 , 36.6) |
| Biceps femoris caput breve | 23.0 (20.7 , 28.1) | 22.4 (20.2 , 28.1) | | | (-34.8 , 24.8) |
| Biceps femoris caput longum | 28.2 (22.7 , 30.2) | 27.5 (25.4 , 32.4) | | | (-16.2 , 29.3) |
| Semimembranosus | 28.7 (21.7 , 32.8) | 28.5 (23.2 , 32.3) | | | (-18.1 , 11.6) |
| Semitendinosus | 32.1 (29.2 , 36.7) | 33.0 (28.3 , 37.1) | | | (-23.6 , 17.7) |
| Rectus femoris | 32.0 (30.8 , 35.4) | 31.8 (30.4 , 34.0) | | | (-86.3 , 16.7) |
| Vastus intermedius | 40.2 (34.8 , 44.5) | 39.0 (36.4 , 42.4) | | | (-7.1 , 22.4) |
| Vastus lateralis | 41.7 (36.8 , 46.2) | 37.6 (34.3 , 40.6) | ** | | (-4.8 , 21.2) |
| Vastus medialis | 37.4 (31.6 , 40.3) | 37.2 (31.2 , 38.9) | | | (-16.0 , 25.7) |
| Tibialis anterior | 32.2 (28.4 , 37.4) | 32.0 (27.3 , 35.2) | | | (-20.5 , 15.8) |
| Tibialis posterior | 31.5 (27.6 , 34.8) | 30.8 (26.9 , 33.5) | | | (-11.4 , 10.2) |
| Gastrocnemius lateralis | 22.5 (19.1 , 28.6) | 24.7 (18.5 , 26.6) | | | (-40.1 , 30.5) |
| Gastrocnemius medialis | 26.9 (25.0 , 33.0) | 27.3 (25.6 , 32.7) | | | (-21.2 , 11.2) |
| Soleus | 36.6 (31.6 , 39.5) | 36.8 (31.8 , 38.9) | | | (-13.8 , 8.1) |
| Peroneus brevis | 25.3 (22.3 , 29.6) | 23.8 (22.1 , 34.6) | | | (-68.7 , 28.8) |

**Fig 2. Muscle length variability.** Median (minimum, maximum) of muscle length for the right and left limb (significant difference between limbs: \*\*p<0.01). Individual percentage difference between the limbs is reported as a bar plot where each bar represents a participant: blue positive (red negative) values show that the right leg is bigger (smaller). Minimum and maximum percentage difference across the subjects is reported for each muscle.

The values found for the participants in this study were similar to those reported by other authors for mixed-age/sex cadavers [25,27], but smaller than those from healthy young adults of both sexes [24] (Fig 5).

## Maximal isometric force

The $F_{max}$ calculated from the VLS approach differed from that of the LLMS by up to 400% (Biceps femoris short head) for individual subjects (Fig 6), with overall smaller estimates of $F_{max}$ with the VLS model. On average, the percentage difference between the two approaches was between -176% (for the Iliacus where $F_{max}$ was smaller in the VLS) and 36% (for the Adductor magnus II where $F_{max}$ was bigger in the VLS). Differences were found significant for all muscles except for Gluteus maximus I and III, Adductor magnus III, Biceps femoris long head, Semimembranosus, Rectus femoris, and Peroneus brevis.

## Discussion

This study aimed to quantify lower-limb muscle anatomical characteristics from medical images in a group of post-menopausal women. To this purpose, the 3D geometries of 23 lower-limb muscles segmented from MRI from a cohort of eleven post-menopausal women were used to assess inter- and intra-individual differences and compared to existing literature data. The use of image segmentation for the calculation of muscle parameters is complicated by the time and repeatability challenges associated with this technique. However, broadening the knowledge of muscle anatomical characteristics could support the development of tools (e.g. population-based statistical models [28] or regression models [24]) to overcome the need for segmentation.

**Table 2. Comparison of muscle volume to literature values.**

| | $V_{MRI}$ | | | $V_D$ | |
| --- | --- | --- | --- | --- | --- |
| | **This study** | **Handsfield et al.** | **Charles et al.** | | **Ward et al.†** |
| Participants* | 22 females | 8 females | 2:1 males:females | | 9:12 males:females |
| Type of study | *in-vivo* MRI | *in-vivo* MRI | *ex-vivo* MRI | dissection | dissection |
| **Muscle** | Mean (SD) | Mean (SD) | Mean (SD) | Mean (SD) | Mean (SD) |
| Adductor brevis | 58.9 (10.6) | 91.5 (16.4) | 79.9 (13.4) | 50.8 (11.2) | 51.7 (23.5) |
| Adductor longus | 78.5 (13.4) | 143.0 (32.5) | 120.3 (37.9) | 97.5 (15.8) | 70.7 (26.9) |
| Adductor magnus | 345.2 (62.7) | 468.8 (86.1) | 511.7 (97.6) | 564.1 (35.2) | 307.5 (121.0) |
| Biceps femoris long head | 110.5 (22.9) | 195.6 (36.9) | 187.5 (51.9) | 200.1 (63.2) | 107.4 (45.9) |
| Biceps femoris short head | 56.1 (15.9) | 78.3 (20.8) | 106.1 (41.5) | 123.4 (63.9) | 56.6 (21.4) |
| Gastrocnemius lateralis | 75.4 (15.2) | 134.9 (17.6) | 140.8 (22.4) | 161.0 (21.7) | 58.9 (23.3) |
| Gastrocnemius medialis | 160.6 (29.3) | 242.8 (36.8) | 245.9 (30.9) | 264.5 (37.7) | 107.5 (30.3) |
| Gluteus maximus | 576.4 (117.8) | 747.3 (92.7) | - | - | 518.2 (153.6) |
| Gluteus medius | 246.2 (58.2) | 286.1 (25.2) | - | - | 259.0 (72.8) |
| Gracilis | 45.2 (14.4) | 88.5 (17.6) | 111.4 (17.6) | 118.1 (8.6) | 49.7 (15.8) |
| Iliacus | 113.8 (17.3) | 150.8 (21.3) | - | - | 107.7 (35.0) |
| Peroneus brevis | 34.8 (8.0) | - | - | - | 22.9 (10.0) |
| Rectus femoris | 120.3 (20.8) | 216.5 (28.5) | 218.8 (42.2) | 235.5 (37.4) | 104.7 (41.0) |
| Sartorius | 83.0 (29.6) | 123.1 (15.2) | 149.3 (26.1) | 165.4 (15.3) | 74.3 (29.5) |
| Semimembranosus | 123.9 (24.0) | 218.8 (34.3) | 220.3 (83.2) | 215.6 (107.1) | 127.2 (54.5) |
| Semitendinosus | 102.5 (26.7) | 146.4 (26.6) | 178.0 (26.5) | 176.5 (31.9) | 94.4 (35.8) |
| Soleus | 339.6 (70.2) | 409.8 (70.4) | 405.6 (143.4) | 437.2 (190.4) | 51.7 (23.5) |
| Tensor fasciae latae | 41.6 (12.8) | 50.3 (21.6) | - | - | - |
| Tibialis anterior | 94.7 (15.3) | 120.6 (22.2) | 151.8 (26.8) | 156.9 (30.0) | 261.2 (93.3) |
| Tibialis posterior | 77.4 (14.9) | 94.8 (12.0) | - | - | 75.9 (25.2) |
| Vastus intermedius | 266.9 (38.5) | 230.6 (46.6) | 360.8 (77.3) | 312.5 (95.9) | 55.3 (18.2) |
| Vastus lateralis | 317.9 (59.2) | 699.1 (102.0) | 691.9 (294.7) | 691.0 (224.2) | 162.8 (69.0) |
| Vastus medialis | 213.8 (29.7) | 354.6 (45.4) | 452.7 (96.8) | 513.3 (95.6) | 356.0 (129.9) |

Mean (standard deviation, SD) of the muscle volume for a subset of lower-limb muscles for the eleven subjects in the current study. $V_{MRI}$ (grey columns) are the volumes obtained from MRI segmentation calculated in our study, in Handsfield et al. [24] from eight healthy young females (30 (8) y.o.), and in Charles et al. [27] from three cadavers (36 (14) y.o.). $V_D$ are the muscle volumes obtained from cadaver dissection from three cadavers by Charles et al. [27] and from twenty-one elderly cadavers (83 (9) y.o.) by Ward et al. [25].

* with participants we here refer to the number of limbs considered independently; gender of the participants is reported too.

† muscle mass values reported by Ward et al. [25] were multiplied by a muscle density of 1.056 g/cm$^3$ as suggested by the authors in [36].

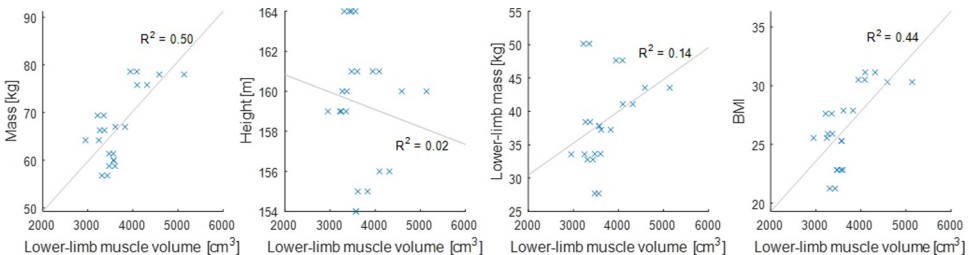

**Fig 3. Linear regression between muscle volume and anthropometric parameters.** Linear regression and coefficients of determination ($R^2$) between total lower-limb muscle volume and body mass ($R^2 = 0.50$, p = 0.003, left), height ($R^2 = 0.02$, p > 0.05, middle-left), lower-limb mass ($R^2 = 0.14$, p > 0.05, middle-right), BMI ($R^2 = 0.44$, p = 0.004, right).

| Muscle | PCSA [cm²] Left | Right | Inter-limb difference [%] Right – Left | (Min , Max) |
|---|---|---|---|---|
| Adductor brevis | 4.0 (2.7 , 6.2) | 4.5 (2.6 , 5.2) | | (-24.1 , 18.8) |
| Adductor longus | 6.0 (4.8 , 7.6) | 5.9 (4.5 , 8.4) | | (-31.8 , 15.4) |
| Adductor magnus | 23.0 (19.1 , 32.4) | 26.1 (22.3 , 32.9) | | (-17.5 , 20.8) |
| Biceps femoris long head | 14.8 (7.8 , 17.3) | 14.5 (11.2 , 16.8) | | (-13.6 , 30.4) |
| Biceps femoris short head | 4.7 (3.1 , 7.0) | 5.1 (2.7 , 7.5) | | (-28.1 , 28.2) |
| Gastrocnemius lateralis | 11.6 (8.4 , 14.1) | 12.4 (9.3 , 15.3) | | (-33.7 , 24.0) |
| Gastrocnemius medialis | 28.6 (26.1 , 36.6) | 27.2 (25.8 , 36.3) | | (-17.8 , 6.8) |
| Gluteus maximus | 27.3 (20.9 , 34.7) | 27.4 (21.4 , 35.5) | | (-4.1 , 14.2) |
| Gluteus medius | 23.3 (17.5 , 34.4) | 23.8 (17.7 , 39.1) | | (-16.8 , 13.4) |
| Gracilis | 1.6 (0.9 , 2.6) | 1.5 (0.6 , 2.8) | | (-83.9 , 37.7) |
| Iliacus | 5.8 (5.4 , 7.1) | 6.1 (4.5 , 8.6) | | (-20.2 , 17.1) |
| Peroneus brevis | 7.3 (4.4 , 8.7) | 7.4 (5.7 , 9.4) | | (-44.5 , 29.4) |
| Rectus femoris | 18.0 (14.4 , 23.6) | 17.5 (10.4 , 23.0) | | (-84.0 , 14.6) |
| Sartorius | 1.5 (1.0 , 2.9) | 1.6 (1.0 , 3.2) | | (-12.5 , 31.4) |
| Semimembranosus | 17.4 (13.3 , 23.0) | 17.8 (13.7 , 23.4) | | (-21.6 , 17.4) |
| Semitendinosus | 4.9 (2.4 , 7.2) | 4.7 (2.8 , 7.2) | | (-25.2 , 21.4) |
| Soleus | 81.2 (69.6 , 134.5) | 83.2 (67.0 , 125.1) | | (-9.1 , 6.6) |
| Tensor fasciae latae* | 2.0 (1.3 , 3.2) | 2.3 (1.0 , 3.2) | | (-30.8 , 32.9) |
| Tibialis anterior | 10.8 (8.8 , 13.1) | 11.2 (8.8 , 16.0) | | (-17.1 , 24.7) |
| Tibialis posterior | 20.6 (15.8 , 32.9) | 21.0 (14.7 , 29.0) | | (-20.6 , 15.4) |
| Vastus intermedius | 26.7 (21.1 , 38.2) | 27.9 (22.6 , 38.2) | | (-25.0 , 23.5) |
| Vastus lateralis | 21.1 (16.3 , 26.5) | 20.3 (14.0 , 30.4) | | (-29.4 , 13.1) |
| Vastus medialis | 26.0 (21.3 , 31.6) | 27.2 (22.6 , 33.5) | | (-12.9 , 19.3) |

**Fig 4. PCSA variability.** Median (minimum, maximum) of the physiological cross-sectional area (PCSA) for 23 lower-limb muscles for eleven subjects in our study (n = number of limbs). PCSA are derived from the segmented $V_M$ and $l_M$ and using the average optimal fibre length to muscle length ratio proposed by Ward et al. [25]; *PCSA of the Tensor fasciae latae was calculated setting the optimal fibre length to muscle length ratio equal to 1 (as proposed by Handsfield et al. [24]) since the actual values were not available from the literature source. Minimum and maximum percentage difference across the subjects is reported for each muscle.

This is, to our knowledge, the first study providing a quantification of lower-limb muscle volumes and lengths in older women, and a thorough assessment of the differences observable both between body sides and across individuals. An ultrasound-based study quantified up to 24% of muscle thickness asymmetry in abdominal muscles in healthy individuals of different ages [37], suggesting that analogous results could be expected in the lower limbs. When comparing the two limbs of each subject in our cohort, we observed differences of up to 85% for $V_M$ and of up to 22% for $l_M$ (Figs 1 and 2). Except for very few muscles (Sartorius, Gluteus maximus, Adductor magnus, Vastus lateralis) which were significantly bigger on the right side, no recognisable pattern was observed across the cohort to be associated with limb dominance. In fact, both muscle volumes and lengths were notably variable in the population. This clearly indicates that care should be taken in assuming limb symmetry when assigning musculotendon parameters, even in healthy populations.

Even though different approaches to the image segmentation may have affected the estimate of the muscle parameters, the comparison to MRI-based values from the literature [24] led to valuable insights. Despite the average height and weight of our participants being smaller than those previously reported for an *ex-vivo* cohort [25], slightly larger $V_M$ were found (Table 2). This could be explained by the loss in muscle mass in cadavers [36]. On the contrary, our $V_M$ was smaller than that estimated *in-vivo* from MRI in healthy young adults (25.5 (11.1) y. o.) [24], which might be explained by both younger age and mixed-sex participants. In fact, when isolating the female component from the young population, smaller average $V_M$ and SD were still observed in our cohort for all the muscles. This explains the smaller inter-subject variability (as quantified by CoV) found in our study, i.e. between 14% (Vastus medialis) and 36%

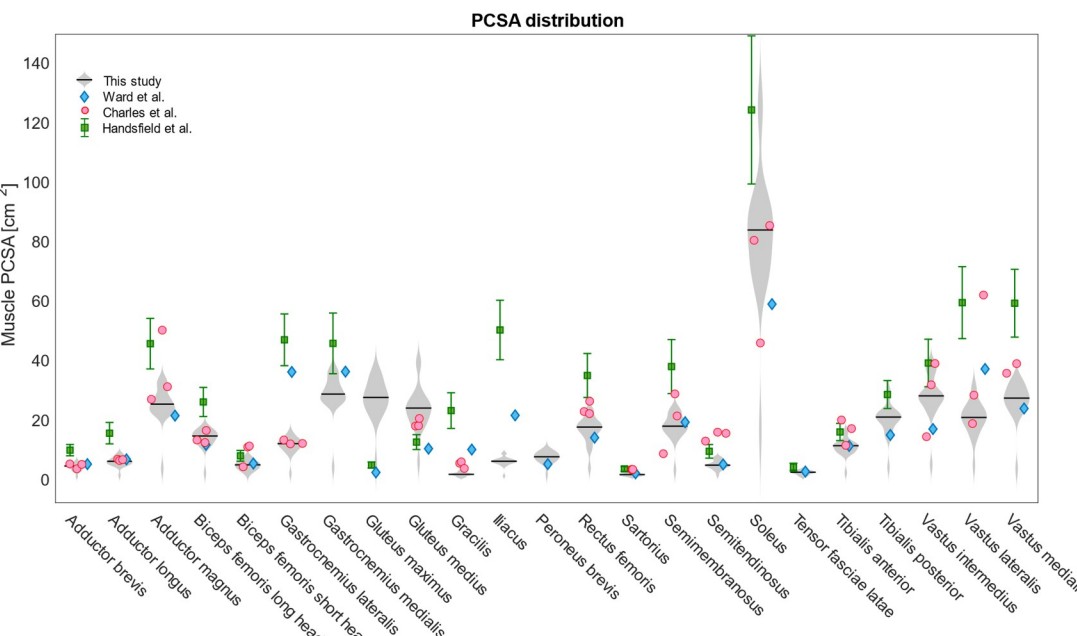

**Fig 5. PCSA distribution and comparison to literature data.** Distribution of the PCSA for the 22 limbs analysed in this study (grey violin plots) compared to PCSA values from literature. Red circles represent individual data points for three cadavers as calculated by Charles et al. [27]. Blue diamonds represent mean PCSA values for twenty-one cadavers as calculated by Ward et al. [25] and divided by the cosine of the mean pennation angle reported by the same authors. Green squares with deviation error bars represent PCSA values estimated by Handfield et al. [24] from MRI segmentation of thirty-two healthy young adults.

(Sartorius), compared to literature values for healthy young mixed-sex adults (quantified between 20% and 40% from the reported mean and SD) [25] and even more when isolating the female component (except for the Tensor fasciae latae muscle). The $V_M$ calculated from our cohort remained consistently smaller to those from young females, except for the Vastus intermedius, likely due to ageing-related muscles volume loss [3,21].

In order to overcome the need for individual muscle segmentations to estimate muscle-specific parameters in MSK models, Handsfield et al. [24] proposed a series of regression equations linearly correlating individual muscle volume to participants' total lower-limb muscle volume, body mass and height. Lower correlations were quantified in this study (Fig 3), likely due to having included only 23 instead of 35 lower-limb muscles. This discrepancy could also be preferential weakening or atrophy of certain muscles caused by ageing [38], an hypothesis which seems to be confirmed by the lower volumes found in our cohort when compared to younger females. Surprisingly, $V_{TOT}$ correlated more strongly with total body mass than with lower-limb mass, suggesting that scaling muscle forces based on lower-limb mass (LLMS) [32,39] might not be a suitable approach in an older population, and a simple scaling to body mass should be preferred in the absence of MRI.

The maximal force that a muscle can produce is highly affected by its PCSA [26]. Since optimal fibre length could not be calculated from available MRI data, the PCSA was here calculated by scaling $l_M$ according to *ex-vivo* literature values from an older population [25]. This led to PCSA values in agreement with literature [24,25,27], except for bigger values for the Gluteus maximus and smaller values for the Iliacus (Fig 5). The PCSA of the Sartorius muscle presented a 37% of CoV between the subjects, due to high variability in its volume and small variability in its length. This was also the only muscle showing significantly different $F_{max}$ between the body sides at group level, with larger values in the dominant limb. Previous studies

**Fig 6. Maximal isometric force calculated with the VLS and LLMS approach.** Median (minimum, maximum) of the maximal isometric force for the VLS and LLMS approaches with p values representing the statistical significance of Wilcoxon test. Individual percentage difference of $F_{max}$ between VLS and LLMS reported as a bar plot where each bar represents a participant: green positive (orange negative) bars show that the value is bigger (smaller) with the VLS approach. Minimum and maximum percentage difference across the subjects is reported for each muscle. * These values correspond to one third of the total muscle $F_{max}$.

highlighted intra-subject variability in the tendon-to-muscle belly length ratio as well as in the location of the widest part of the muscle along its axis [40], therefore confirming our findings.

The specific tension (σ) of a muscle also contributes to the estimate of $F_{max}$. The choice of setting σ to 61 N/cm$^2$ was suggested by previous literature where this value was proposed for elderly populations [16,35]. Sensitivity of models to this parameter was previously tested by Valente et al. [13], finding a moderate effect on the model output. In the effort of maximally personalizing muscle parameters, individual values for the specific tension should be obtained for different subjects and different muscles, however such a measure is not currently available *in-vivo*. The use of dynamometer could provide further insight in the specific tension of muscle groups and overcome this limitation.

Estimated $F_{max}$ were overall significantly smaller when based on $V_M$ than when linearly scaled to lower-limb mass (Fig 6), except for the Adductor magnus, Vastus intermedius and medialis, Gastrocnemius medialis, and Soleus, that, on the contrary, presented significantly higher values for the LLMS approach. Declining muscle strength has been observed from the age of fifty [3] and a reduction by 20% of $F_{max}$ has been quantified in older people aged seventy [41]. This could explain the smaller $F_{max}$ obtained from individual $V_M$ (when volume loss associated with ageing was taken into account) compared to a scaling approach. This also confirms previous literature suggesting that a scaling approach might only be appropriate if starting from values from a sex- and age-matched population [28].

The choice of $F_{max}$ highly impacts the output of MSK models [42,43], since a change in an individual muscle ability to produce force alters the solution of the static optimisation problem

[42], affecting both individual muscle force estimates and the resulting joint contact force. A previous study found limited sensitivity of muscle forces and joint contact forces to $F_{max}$ [32] estimating its values based on scaling of literature values or using Handsfield's regression equations. Ackland et al. [19] studied the effect of variation between +10% and −10% of $F_{max}$ nominal value, reporting no significant changes in the model output. However, in their study, they did not account for actual muscle geometry to estimate $F_{max}$, which proved to cause variation up to 400% in our study when compared to scaling approaches. This suggests that calculating individual $F_{max}$ from MRI-segmentations could affect the estimates of muscle forces and joint contact forces on a larger scale than reported in the literature and lead to more accurate estimates. This supports the conclusions from Arnold et al. [16] that tuning individual muscle parameters might provide estimates of internal forces that compare better to experimental measurements [16]. Further studies are needed to confirm this hypothesis.

This study had some limitations. Out of the 35 muscles commonly included in lower-limb MSK models, only 23 were included in the study, as these were not significantly affected by operator-related error in the segmentation. Muscle segmentation is a time-consuming (10 hours per subject on average for this study) and operator-dependent procedure, therefore further effort should be put into developing automated algorithms based on machine learning [28] for the segmentation of individual muscles or statistical shape modelling-based approaches for the extraction of muscle volume and muscle centreline/length. The dataset associated with this paper is publicly available, which will likely foster advances in this field, i.e. acting as a reference atlas.

The cohort enrolled for this study included eleven participants; a larger sample size would be needed to ensure generalisability of the results observed here. Our results suggest that muscle asymmetry could be higher in older adults due to age-related processes. However, this finding is based on comparison to literature [24], where data were obtained following a slightly different methodology. Therefore, a wider study, including a control group of younger women, should be designed to prove our hypothesis.

In the attempt of preserving a degree of subject-specificity in the muscle parameters, PSCA was calculated from muscle volume and length. Nonetheless, due to the impossibility of estimating the optimal fibre length from the implemented MRI sequence, the required ratio between optimal fibre length and muscle length was taken from cadaveric data. Diffusion Tensor imaging recently proved to be a valuable option to enable both muscle segmentation and the estimate of fibre length [27]. Further studies are needed to understand whether this technique might be included within an MSK modelling imaging protocol to overcome this limitation.

In conclusion, this study uniquely proved the existence of significantly large muscle- and subject-specific asymmetry in muscle volume, length, and PCSA. This suggests that individual differences in muscle geometry must not be neglected, and inter-limb symmetry cannot be assumed in older women. Personalised muscle characteristics should be accounted for in MSK models aiming at investigating dynamic tasks such as walking, where strength asymmetry plays an important role in older women. This could be of substantial relevance when internal forces are used in clinical contexts, such as prediction of osteoporotic risk of fracture.

## Supporting information

**S1 Table. Anthropometric data.**
(DOCX)

**S2 Table. Right-limb muscle volumes segmented by three operators for three randomly selected subjects.** Maximum coefficient of variation (CoV) across the three datasets is

reported.
(DOCX)

**S3 Table. Right-limb muscle volumes segmented three times by one operator.** Coefficient of variation (CoV) across the three repetitions is reported.
(DOCX)

**S4 Table. Right and left volume of the muscles segmented in the lower limbs of the eleven subjects enrolled in the study Mean, standard deviation (SD) and coefficient of variation (CoV) are reported.**
(DOCX)

**S5 Table. Right and left length of the muscles segmented in the lower limbs of the eleven subjects enrolled in the study Mean, standard deviation (SD) and coefficient of variation (CoV) are reported.**
(DOCX)

**S6 Table. Physiological cross-sectional areas (PCSAs) measured for the eleven subjects enrolled in our study (for right and left muscles) and three cadavers included in Charles et al., 2019.** Mean and mean and SD PCSA are reported for Ward et al. 2009 and Handsfield et al. 2014, respectively.
(DOCX)

## Acknowledgments

The authors would like to acknowledge Dr Geoffrey Handsfield for sharing relevant data from [24] and Dr Enrico Dall'Ara for his valuable input around image processing.

We are particularly grateful to the participants who volunteered for the study.

## Author Contributions

**Conceptualization:** Margaret A. Paggiosi, Eugene V. McCloskey, Claudia Mazzà.

**Data curation:** Barbara M. Kalkman, Margaret A. Paggiosi.

**Formal analysis:** Erica Montefiori, Barbara M. Kalkman.

**Funding acquisition:** Eugene V. McCloskey, Claudia Mazzà.

**Investigation:** Eugene V. McCloskey, Claudia Mazzà.

**Methodology:** Erica Montefiori, Barbara M. Kalkman, William H. Henson, Margaret A. Paggiosi, Claudia Mazzà.

**Project administration:** Eugene V. McCloskey, Claudia Mazzà.

**Resources:** Margaret A. Paggiosi, Eugene V. McCloskey, Claudia Mazzà.

**Software:** Erica Montefiori.

**Supervision:** Margaret A. Paggiosi, Eugene V. McCloskey, Claudia Mazzà.

**Validation:** Erica Montefiori.

**Visualization:** Erica Montefiori.

**Writing – original draft:** Erica Montefiori.

**Writing – review & editing:** Erica Montefiori, Barbara M. Kalkman, William H. Henson, Margaret A. Paggiosi, Eugene V. McCloskey, Claudia Mazzà.

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
