## [Decision Letter · Decision Letter 0]

30 Sep 2020

PONE-D-20-23990

MRI-based anatomical characterisation of lower-limb muscles in post-menopausal women

PLOS ONE

Dear Dr. Montefiori,

Thank you for submitting your manuscript to PLOS ONE. After careful consideration, we feel that it has merit but does not fully meet PLOS ONE’s publication criteria as it currently stands. Therefore, we invite you to submit a revised version of the manuscript that addresses the points raised during the review process.

Please see the insightful comments from both reviewers that I feel fall collectively into major rather than minor amendments. Addressing their comments will positively enhance the quality of your manuscript.

We look forward to receiving your revised manuscript.

Kind regards,

Alison Rushton

Academic Editor

PLOS ONE

Reviewers' comments:

Reviewer's Responses to Questions

**Comments to the Author**

1. Is the manuscript technically sound, and do the data support the conclusions?

Reviewer #1: Yes

Reviewer #2: Yes

2. Has the statistical analysis been performed appropriately and rigorously? 

Reviewer #1: Yes

Reviewer #2: Yes

3. Have the authors made all data underlying the findings in their manuscript fully available?

Reviewer #1: Yes

Reviewer #2: Yes

4. Is the manuscript presented in an intelligible fashion and written in standard English?

Reviewer #1: Yes

Reviewer #2: Yes

5. Review Comments to the Author

Reviewer #1: 

The article is technically sound but some correction needs to be carried out to add to the quality of the manuscript. Find below some of my observations regarding the manuscript.

Short title: this should be a short form not a replica of the full title.

Abstract

Line 29 “These variables were compared …” No need of this statement here. It looks like repetition of your aim “… their variability in a group of post-menopausal women”

Line 34 “Personalised maximal isometric force was then calculated …” is part of the methods.

Line 36 Your conclusion is not in line with your stated aim in the abstract.

Line 42 “Key words” is one-word, Keywords. Arrange your keywords alphabetically and you may wish to limit the number to 5 to 7, which is enough.

Introduction

Line 84 “form” from

Materials and Methods

Line 92 BMI should be written in full in the first instance. In the exclusion/inclusion criteria, do you consider the exercise history of the patients? If not, this should be stated as part of the limitation in the discussion.

Line 99 “During …” This should be a new paragraph.

Line 115 “Psoas muscle” Psoas major muscle? Or Psoas muscles were

Line 116 “valuated” evaluated

Line 146 You need to state whether your data pass the normality test or not.

Line 147 “MATLAB” version please.

Line 151 Did you employed any statistics (e. g one sample test) in comparing your results with that of literature? If not, this sentence is not relevant here.

Line 152 state the type of correlation e. g Pearson’s or Spearman rank order correlation etc.

Line 153 use the abbreviated form of body mass index here.

Line 158 the significant level should be set at P < 0.05.

Results

Line 161 delete “As expected”

Lines 168 to 182, Table 1 under the body segment, thigh. This should be thigh and gluteal. Body segments and Muscles in plural forms.

Discussion

Lack of control population for young adults may need to be stated as additional limitation of the study. Since it is clear that the subject’s characteristics varied between the present study and compared population in the literature.

Despite the time-consuming nature of the muscle segmentation techniques, larger sample size may be needed to give better picture of the observed results. This may be added as recommendations.

Conclusion

The first few sentences should be based on your results and in line with your objectives. You can conclude on the implications of your study. Avoid citation unless necessary.

Line 355 “Acknowledgment” Acknowledgments

Reviewer #2: 

General impression:

This manuscript investigated anatomical parameters of lower limb muscles based on their segmentations from MRI images and on two different approaches for calculating the maximal isometric force (Fmax) in 11 post-menopausal women. One calculation approach was based on a linear scaling of the lower-limb mass; the second approach was based on individual muscle volumes and length.

By comparing the two calculation methods, the authors found a significant difference in the estimated Fmax for most muscles, reasoning that individually calculated values for each muscle should be preferred to linear scaling methods. Secondly, their findings suggest that a considerable asymmetry regarding volume, length and physiological cross-sectional area (PCSA) might exist between the left and right limb, independently from limb dominance. This is a finding that has not been considered much in musculoskeletal modelling so far, but it can have a great impact on force calculation.

A great strength of this paper is the aim of the authors to improve the quality of scaling methods by making their data publicly available for future research. Also, it is generally well understandable and structured.

Nevertheless, there are some aspects that could benefit from revision. The Methods section is detailed and well written, but the high inter-operator variability in the segmentation procedure raises the question of how reproducible this method is. The authors should comment on this in the discussion. The figures represent the main findings of the results section, however, some figure legends should be extended to explain the figure more completely. Furthermore, the authors should pay attention to homogenize the use of terms in the text, figures and figure legends, as this would support the comprehension of the results. Also, we would recommend the manuscript to be copyedited regarding syntax and spelling errors. Since our expertise in the field of statistics is limited, we do not comment much on this topic; nonetheless, the chosen tests seem to be reasonable for the respective aims. We would highly appreciate, if the MATLAB script could be made publicly available, e.g. on Figshare or GitHub.

Even though each of the recommended revisions are minor, in total there is still substantial work to be done on the manuscript to increase its quality before publishing.

Specific comments:

Abstract:

- Line 23: “The ability of muscles to produce force depends on”, not by their anatomical features. Do not use shortages (it’s).

- Line 29: “between left and right limbs”

- Line 32-33: “Generally, muscle parameters were similar … but volumes were smaller”, otherwise we would expect the volumes to be similar, too.

Introduction:

- Line 72: it is not clear what “the ratio of which” refers to, please clarify this sentence.

- Line 84: from instead of form.

Methods:

- The authors should explain why they chose women with osteopenia or osteoporosis (line 89-92) and mention it both in the abstract and the discussion. Do the authors expect any possible impacts on their results or not and if yes, what could they be like?

- Line 130-135: In our opinion the muscle length could have been calculated with a 3D extraction of the centreline instead of extrapolating it from filtered centrelines calculated on the 2D slices. Why was the muscle length not calculated directly from the 3D data? For example, with https://www.mathworks.com/matlabcentral/fileexchange/43400-skeleton3d or https://www.mathworks.com/matlabcentral/fileexchange/71766

- The section about intra- and inter-operator repeatability should be clarified. Under which criteria was the intra-operator variability performed additionally and which approach was used for which muscle? This could for example be indicated in Table 1. Why was the cut-off for exclusion of a muscle set at 10%?

- The ± sign should be specified in the text; does it mean standard deviation (SD) or range? Or in which case does it mean what? For example, line 88 about the age, weight and height of the subjects: as readers, we assume this to be the range, however, compared to the numbers in Fig. 3, it does not seem to be the case.

- Line 150: how was the CoV calculated?

- The abbreviation of “muscle volumes and length” is not the same throughout the text (line 122: VLS, line 130: MVLS). The authors should take care to always use the same abbreviation for the same term.

- A citation or details for the MultiSim Study (line 89) should be given.

- Line 91: the abbreviation DXA is not necessary, since the term appears only once in the whole manuscript.

Results/Figures:

- Fig. 1, 2, and 4: switching column 2 and 3 (“Right” and “Left”) might increase the readability of the figures, since the inter-limb difference in % (column 4) is presented in this way. The authors should use the same notation for range in the whole figure and all the figures (either with a comma or a hyphen, if possible, also not use square brackets). In the text of the results the authors mention the between-subject CoV (line 190 for the muscle volume and line 192 for the muscle length), but they do not state which numbers belong to the volume and which to the length. Furthermore, the numbers for the between-subject CoV from the text are not derivable from the numbers in the figures, the authors should clearly indicate which column shows the between-subject CoV in the figures.

- Figure 6: what does the p value in column 4 refer to? Does it show significance in the difference between LLMS and VLS? Then this should be mentioned in the figure legend and in the methods section, where the authors only state that the Wilcoxon test was used to show significance in limb asymmetry. Adjusting the presentation of the p values according to the other figures would be favourable (using * and ** instead of providing numbers). Furthermore, the discussion of those results (line 315-316) should be clarified. If we understand it right, the muscles mentioned there are not the muscles with no significant difference, but the ones with higher values calculated with the VLS approach.

- Table 2 (line 213) could be better understandable if the authors would indicate clearly which study was based on living subjects or cadavers and which used MRI or dissection. Why is the study by Ward et al. the only one without SD? Since the numbers seem to have been calculated by the authors (on a quick glance they do not seem to appear in the original work by Ward et al.), it would be nice if they could also provide the SD. Also, using Vm for muscle volumes generated specifically by MRI (as opposed to dissection) in this table is confusing since it usually stands for muscle volume.

- Line 187/188: the sentence sounds as if the percentage differences of Vm in the Gracilis and Rectus femoris were higher than 85% in all the subjects, however it is the case only in one subject for each of the muscles. The authors should rephrase this sentence.

- Fig. 3 legend: Please complete the text according to the results, describing correlations and indicating significant values. Correct lower-limb mass (line 222).

- Fig. 5 legend: what does the figure show? Please complete the text. Correct PCSA values (line 241).

- Fig. 5: Separating the data points from the three presented studies with a jitter or separation would increase the readability. Furthermore, it would be useful to indicate whether data points represent single subjects or mean values.

- Line 224: “(fig. 4)” needs to be formatted consistently with other figure mentions.

Discussion:

- In order to follow the flow of the discussion it might be useful to adhere to the order of the results and figures and to reference the figure which shows the respective results.

- Since the calculation of Vtot consisted of only 23 muscles, the authors should discuss this as an additional possible reason why it did not correlate with the lower-limb-mass.

- It would be interesting to know if there is a possible explanation for the asymmetric left/right distribution of the muscle volume and length. Is there any recognisable pattern, e.g. that some individuals are in fact rather left- than right-footed, or could it be the case that smaller muscles in one functional group are being compensated by larger volumes of the other muscles in the same group? Could additional anamnestic info about hip/knee problems or movement limitations be helpful to understand possible patterns?

- Line 328: “which proved to cause variation up to 400% in our study …"

- Grammar corrections: line 264 “However, …"; line 301 “Valente et al., finding ...”; line 306 “In order to overcome …”

Acknowledgement:

- Line 356: from instead of form; line 358: grant numbers (since they are two); line 360: we assume that the “the views expressed” are the ones of all the authors, hence author(s) should be corrected.

6. PLOS authors have the option to publish the peer review history of their article (what does this mean?). If published, this will include your full peer review and any attached files.

Reviewer #1: **Yes: **Dr Lawan Hassan Adamu

Reviewer #2: **Yes: **Dea Aaldijk

---

## [Author Response · Author response to Decision Letter 0]

21 Oct 2020

We would like to thank both reviewers and editor for their very constructive and accurate feedback. We have now implemented their suggestions, which we feel have helped in greatly improving the paper.

---

## [Decision Letter · Decision Letter 1]

12 Nov 2020

PONE-D-20-23990R1

MRI-based anatomical characterisation of lower-limb muscles in older women

PLOS ONE

Dear Dr. Montefiori,

Thank you for submitting your manuscript to PLOS ONE. After careful consideration, we feel that it has merit but does not fully meet PLOS ONE’s publication criteria as it currently stands. Therefore, we invite you to submit a revised version of the manuscript that addresses the points raised during the review process.

We look forward to receiving your revised manuscript.

Kind regards,

Alison Rushton

Academic Editor

PLOS ONE

Additional Editor Comments (if provided):

Please address the final comments detailed below from reviewer #1.

Reviewers' comments:

Reviewer's Responses to Questions

**Comments to the Author**

1. If the authors have adequately addressed your comments raised in a previous round of review and you feel that this manuscript is now acceptable for publication, you may indicate that here to bypass the “Comments to the Author” section, enter your conflict of interest statement in the “Confidential to Editor” section, and submit your "Accept" recommendation.

Reviewer #1: All comments have been addressed

Reviewer #2: All comments have been addressed

2. Is the manuscript technically sound, and do the data support the conclusions?

Reviewer #1: Yes

Reviewer #2: Yes

3. Has the statistical analysis been performed appropriately and rigorously? 

Reviewer #1: Yes

Reviewer #2: Yes

4. Have the authors made all data underlying the findings in their manuscript fully available?

Reviewer #1: Yes

Reviewer #2: Yes

5. Is the manuscript presented in an intelligible fashion and written in standard English?

Reviewer #1: Yes

Reviewer #2: Yes

6. Review Comments to the Author

Reviewer #1: The authors responses to the comments are satisfactory. Now the work can be accepted for publication but the authors should take note of the following:

Line 28 Lower limb or Lower-limb. Please be consistent

Line 102 “Declaration of Helsinki” Please give year

Line 114 “SD” write in full.

Line “References” need formatting e.g. Journal names are mixed of Title case and Sentence case

Thank you.

Best regards

Reviewer #2: (No Response)

7. PLOS authors have the option to publish the peer review history of their article (what does this mean?). If published, this will include your full peer review and any attached files.

Reviewer #1: **Yes: **Lawan Hassan Adamu

Reviewer #2: **Yes: **Dr. med. Dea Aaldijk

---

## [Author Response · Author response to Decision Letter 1]

12 Nov 2020

We would like to thank the reviewer for this further feedback to our manuscript. We have now implemented the suggestions.

---

## [Editor Report · Decision Letter 2]

13 Nov 2020

MRI-based anatomical characterisation of lower-limb muscles in older women

PONE-D-20-23990R2

Dear Dr. Montefiori,

We’re pleased to inform you that your manuscript has been judged scientifically suitable for publication and will be formally accepted for publication once it meets all outstanding technical requirements.

Kind regards,

Alison Rushton

Academic Editor

PLOS ONE

Additional Editor Comments (optional):

Thank you for addressing the outstanding minor points from reviewers.
---

## [Editor Report · Acceptance letter]

19 Nov 2020

PONE-D-20-23990R2 

MRI-based anatomical characterisation of lower-limb muscles in older women 

Dear Dr. Montefiori:

I'm pleased to inform you that your manuscript has been deemed suitable for publication in PLOS ONE. Congratulations! Your manuscript is now with our production department. 

Kind regards, 

on behalf of

Professor Alison Rushton 

Academic Editor

PLOS ONE